# Home Greenery: Alleviating Anxiety during Lockdowns with Varied Landscape Preferences

Zhengkai Zhang *, Hanjiang Zhang, Huan Yang * and Bingzhi Zhong

College of Landscape Architecture and Art, Northwest A&F University, Yangling 712100, China; hanjiang.z.0302@gmail.com (H.Z.); zhongbingzhi@nwafu.edu.cn (B.Z.)
* Correspondence: zkzhang@nwafu.edu.cn (Z.Z.); yanghuan@nwafu.edu.cn (H.Y.)

**Abstract:** During the COVID-19 pandemic, many countries applied lockdown rules to flatten their epidemic curves. Meanwhile, many people suffered mental health crises. However, evidence is lacking on the psychologically restorative effects of home greenery for citizens with varying landscape preferences when public green spaces are unavailable. In Xi'an, China, during the December 2021 lockdown period, a questionnaire on residents' anxiety, houseplants and green view from windows, and landscape preferences was designed by the authors and sampled by snowballing. Houseplants and green view from windows were positively associated with anxiety remission ($p < 0.05$), and the effects were different among landscape preferences. The houseplants helped to alleviate moderate and severe anxiety among respondents who preferred open green spaces and partly open green spaces. Visual exposure to Urban Green Spaces through windows alleviated mild anxiety in respondents who preferred open green spaces. It also alleviated mild, moderate, and severe anxiety in respondents who preferred partly open green spaces. More visual exposure to Urban Green Spaces via windows alleviated mild, moderate, and severe anxiety in respondents who preferred partly open blue spaces. When cities are at risk of pandemics, or in places where incapacitated people are living, distributing indoor plants to households presents a quick approach to helping mitigate anxiety and increasing green cover in residential areas will improve sustainability.

**Keywords:** green spaces exposure; landscape preferences; lockdown; anxiety relief; COVID-19



## 1. Introduction

During the COVID-19 pandemic, many countries applied lockdown rules in an attempt to flatten their epidemic curves [1,2]. Because of the numerous and unpredictable risks of the pandemic and of these rules confining people to their homes, people have faced social isolation, changes in lifestyle, and psychological problems [3,4]. Evidence has shown that staying at home throughout the pandemic or shielding at all were strongly associated with greater risk of elevated depressive symptoms, anxiety, poorer quality of life and lower life satisfaction.

Green spaces bring a variety of ecosystem services to residents, including regulation services and cultural services [5]. They are considered to be an important environmental factor that affects people's physical and mental health [6]. There are two fundamental theories that argue about the positive effects of exposure to nature on mental health, the Attention Restoration Theory (ART) [7] and the Stress Reduction Theory (SRT) [8]. According to the ART, the natural environment constitutes a situation that does not require directed attention and allows an exhausted person to rest and recover his/her mental exhaustion [7,9,10]. The natural environment makes people feel comfortable and peaceful by keeping one's attention effortlessly, being away from anxious environments and creating a sense of immersion [7,9,11,12]. It also brings many benefits to human health and well-being [6,10,13]. According to the SRT, humans have an innate connection to safety and nature, such as trees and water [8]. Exposure to artificial settings was associated with higher

levels of mental stress, while natural features reduced mental stress, and promoted mental health and well-being [14–20]. Some researchers have found that momentary UGS (Urban Green Spaces) exposure, for example, staying in a green space for 10 min, participating in a program of horticultural therapy, could help people feel better [21,22].

Researchers have developed many indicators to estimate UGS exposure which may be broadly classified as potential or actual exposure [23]. Metrics of potential exposure refer to availability, accessibility, and attractiveness [20,23,24]. Applied in surveys using questionnaires and field studies, metrics of actual exposure include self-reported exposure, perceived access to UGS through window views, and reported visits to UGS for outdoor activities [19,25,26].

Based on the ART theory, indoor plants have a positive effect on restoring people's stress and anxiety. Some studies have used the presence of plants, number of plants, and practice of care for plants to estimate indoor greenery [27–31]. Their results showed that presence of plants and increases in their numbers had positive effects on individuals' statements of physiological well-being [27,28,31]. Some studies have estimated green exposure through windows with reference to the extent of nature viewed from windows or the number of trees out of windows. Their results showed green features viewed through windows could partially alleviate the feeling of missing immersion in UGS [32]. Green features viewed through windows also contributed to well-being and to reduced mental distress [28,29]. Some studies have used birds to estimate green exposure at windows, their results showing that the sight or sound of birds was positively associated with residents' self-reported mental well-being and negatively associated with residents' self-reported anxiety [33,34]. These studies have also found that green features' restorative effects, for those who could not physically access UGS, depended on the frequency of visitation before the pandemic, distance to UGS, generation demographics, and peoples' age [28,29,32].

There is an abundance of evidence showing that people need urban green spaces (UGS) more than before, during the pandemic [11,32,35–38]. Also, residents have increased their use of urban green areas [39]. Lockdown restrictions often set a maximum travel distance from home, so UGS close to residential areas have become more important than before [32]. During some lockdowns, only essential services have been allowed to run at normal capacity, all other services completely closed or restricted—including leisure activities in open spaces.

However, evidence is lacking on the role of green features within and around living environments when public green spaces are inaccessible or unavailable. Researchers have studied the impact of green exposure, indoors and through windows, on people's mental health and well-being during the COVID-19 pandemic. Their studies used questionnaires for psychological screening, such as the PHQ-9, the GAD-7, the POMS and the 21-DASS, and happy feeling scales to identify the effects of green and blue features on people's anxiety, stress, depression, and subjective well-being [11,27–29,31,38,40,41].

Human preferences can be seen as expressions of preference for adaptation to environments that include elements and spaces which support human activity [7]. The object of human preference refers to landscape features' shapes, colors, spatial arrangements, and other visual attributes [42]. The mechanism of landscape preferences can be explained by survival-related hypotheses focusing on such inherent behaviors as hunting and sheltering [43]. Some studies found that people's aesthetic preferences tend to favor more natural, wild landscapes with flowers and more trees [44–48]. In addition, landscape preferences were associated with psychological restoration [49,50]. Studies employed landscape configuration and openness to categorize respondents' landscape preferences and used their mood states and a perceived restorative scale to quantify the psychological restoration of UGS [45,47,51–55]. Their results showed revealed a positive association between eye-level tree cover density, perceived naturalness and people's self-reported stress reduction [45,47,49,55,56]. The partly open UGS have the strongest restorative effects on low moods, and the closed UGS have the weakest restorative effects [52].

However, the role of landscape preference in the psychological restoration by UGS has rarely been mentioned, especially during the COVID-19 pandemic. In response to this issue, the aim of this research was to explore the psychologically restorative effects of indoor UGS exposure for people with different landscape preferences when public green spaces were inaccessible during a COVID-19 lockdown. This was carried out using online questionnaires about UGS exposure, landscape preferences, and psychological status of Xi'an residents, during the strict lockdown period. This study provides a basis for UGS planning in residential areas, helps to improve UGS effects alleviating residents' anxiety, and aids in promoting urban sustainability to meet the challenges of future pandemics.

## 2. Materials and Methods

### 2.1. Participants and Study Design

Xi'an is the capital of Shaanxi Province and has one of the largest populations and city sizes in western China (Figure 1). In this city, the ratio of coniferous trees to deciduous trees is about 4:6 [57]. In December 2021, Xi'an reported its first case of COVID-19 (delta), which then increased rapidly and spread across the city. Like in other cities that faced serious threats from the pandemic, to prevent the epidemic from spreading further, the government of Xi'an began to take lockdown measures for the whole city beginning on 23 December. On 24 January, the number of new daily cases was back to zero. The lockdown lasted a total of 32 days. During this lockdown period, most non-essential activities were suspended in the built-up area of Xi'an. Factories were shut down, teaching was conducted online, residents stayed at home, and venues including open spaces such as parks and squares were closed to the public.

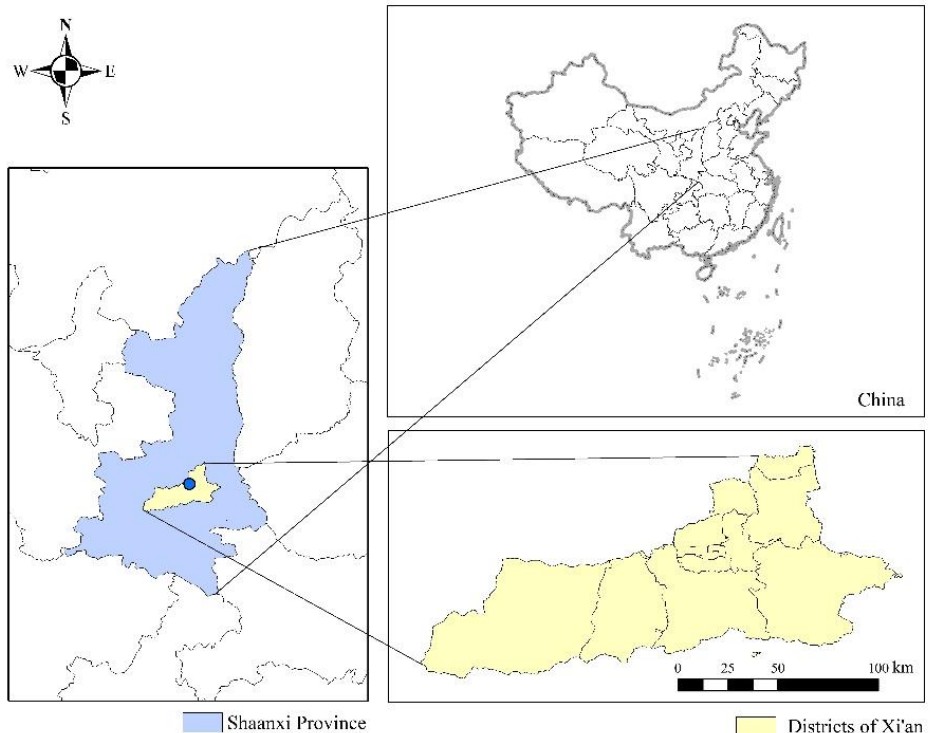

**Figure 1.** Study area.

From 6–10 January 2022, we distributed online questionnaires to 1331 respondents on the survey platform www.wjx.cn. We snowballed the sample by first sending questionnaires to friends who lived in Xi'an. We limited respondents' IP addresses to Xi'an to ensure the spatial range of the sample. The questionnaire data were collected over a period of two weeks. The collected questionnaires were screened by answer time (>90 s) to obtain valid samples.

### 2.2. Measures

Variables for measurement in this study included exposure to green, psychological statements, landscape preferences, the forms of living unit and respondents' demographic variables.

### 2.2.1. Exposure to Home Greenery

Our study used self-reported exposure to assess home greenery exposure. It included five variables: the number of green potted plants at home (hereinafter referred to as "number of houseplants"); the frequency of looking out the window each day ("frequency of looking out the window"); the presence of green space, water body or garden outside the windows, the maximum number of trees that can be seen outside the window ("window green view"); the frequency of bird tweets heard during the lockdown ("heard birdsong"); and the frequency of visits to green spaces before the lockdown ("habits of using green space").

The number of houseplants was assessed using the following question: "How many potted plants are there in your house?" Respondents were asked to choose one of these answers: "zero", "one or two pots of plants", or "more than three pots of plants".

Frequency of looking out the window was assessed using the following question: "During this lockdown period, how often did you look out the window every day?" Respondents should choose one of these answers: "less than once a day", "occasionally (once or twice a day)", "often (more than three times a day)".

The presence of green space, water body or garden outside the windows was assessed using the following question: "Is there any green space, water body or garden outside your windows?"

Window green view was assessed using the following question: "What is the maximum number of trees you can see from the windows of your house?" Respondents should choose one of these answers: "almost none", "a few (1–2)", "many (3 or continuous growth)."

Heard birdsong was assessed using the following question: "How often did you hear birdsong during the lockdown?" Respondents should choose one of these answers: "never noticed", "barely heard", "occasionally heard", "often heard," "heard almost every day".

Habits of using green space were assessed by the following question: "How often did you visit parks or other green spaces before this lockdown?" Respondents should choose one of these answers: "once a month or less", "once or twice a month", "once or twice a week", or "more than three times a week".

### 2.2.2. Psychological Statement

The psychological statement was assessed using the Generalized Anxiety Disorder 7-item (GAD-7) scale. It was designed to assess how often a person is troubled by common symptoms of anxiety, including feeling nervous, worrying too much, having trouble relaxing, becoming easily annoyed, and feeling afraid that something bad might happen [58]. Response options ranged from 0 (not at all) to 3 (nearly every day). The total score (sum of all the item responses) could range from 0 to 21.

### 2.2.3. Landscape Preferences

Many studies on landscape perception and preference indicated that people's perceived value of green space can be divided into certain characteristics, and these characteristics largely reflect people's needs for different types of green space [59–61]. Based on the openness and composition of UGS [52], five types of landscapes were set up to assess residents' landscape preferences, including open green space, partly open green space, closed green space, partly blue space, and partly open square (Figure 2).

### 2.2.4. The Forms of Living Unit

The floor and orientation were used to represent the form of each respondent's living unit. The floor was assessed using the following question: "Which floor do you live on?" Answers included "floors one to three", "floors four to six", "floors six to eleven", and "higher than twelve floors". The unit orientation was assessed using the following ques-

tion: "Which of the following descriptions of housing matches your home?" Respondents should choose one of these answers: "My apartment has south-facing windows"; "My apartment has no south-facing windows, but it has east-facing or west-facing windows"; "My apartment has only north-facing windows".

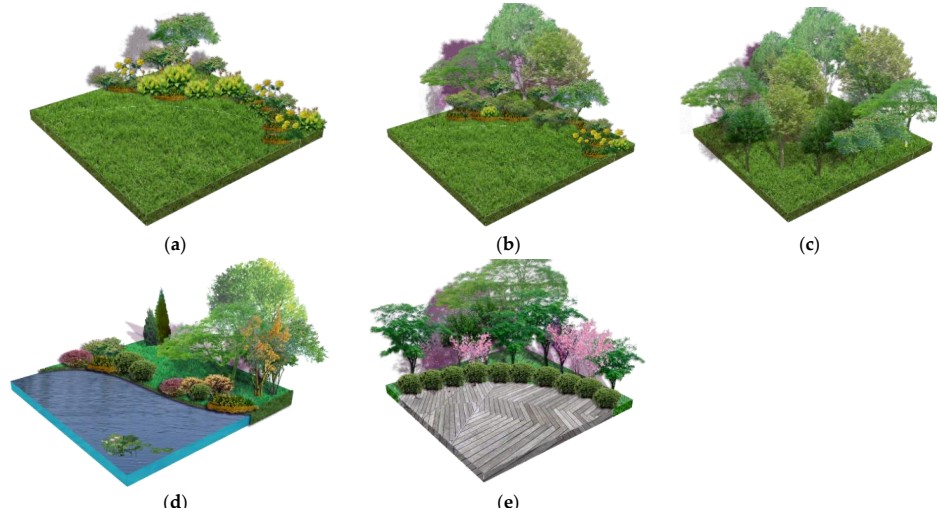

**Figure 2.** Landscape preferences in the questionnaire. (**a**) Open green spaces, (**b**) partly open green spaces, (**c**) closed green spaces, (**d**) partly open blue spaces, and (**e**) partly open squares.

2.2.5. Demographic Variables

The questionnaire included three demographic variables: age, gender, and the district of city (Figure 1).

*2.3. Statistical Analysis*

To examine the autocorrelation of the independent variables, we first conducted a correlation analysis between pairs of all variables except mental state. We then used multinomial logistic regression analysis to test the psychological restorative effects of green exposure during the lockdown. The dependent variable for both regression models was anxiety level. According to the sample size of the four anxiety levels on GAD-7 [58], we reclassified them into three levels to ensure the balance of the sample size in the analysis (Table 1). Regression 1 was to investigate how possible influencing factors may explain variations in psychological statement. The independent variables are shown in Table 2. After that, we used multinomial logistic regression analysis in different landscape preference groups to find out whether the anxiety remission effects were different in these groups (Regression 2). Because the effect of the presence of green space, water body, or garden was covered by other green exposure variables in the first multinomial logistic regression (Table 3), it would not join Regression 2. Since the sample size of the partly open square group was lower than 10% of the largest group (partly open blue space), it was not added in Regression 2 (Table 3). All statistical analyses were run on an IBM SPSS 26.0 (IBM, Armonk, NY, USA).

**Table 1.** The reclassification of anxiety levels.

| Score of GAD-7 | Anxiety Level in Four Class [58] | Reclassified Anxiety Level | Sample Size |
|---|---|---|---|
| 0–4 | minimal | minimal | 821 |
| 5–9 | mild | mild | 321 |
| 10–14 | moderate | moderate and severe | 96 |
| 15–21 | severe | | 58 |

**Table 2.** The independent variable of two multinomial logistic regression analysis.

| Independent Variable | Data Type | Regression 1 | Regression 2 |
|---|---|---|---|
| Gender | categorical | ● | ● |
| Age | categorical | ● | ● |
| Unit floor | categorical | ● | ● |
| Unit orientation | categorical | ● | ● |
| The presence of green space, water body or garden | categorical | ● | ○ |
| Number of houseplants | categorical | ● | ● |
| Frequency of looking out of the windows | categorical | ● | ● |
| Green views through windows | categorical | ● | ● |
| Heard birdsong | categorical | ● | ● |
| Habits of using green spaces | categorical | ● | ● |

●: The variables that input in the regression model. ○: The variables that are filtered out.

**Table 3.** The demographic characteristics and landscape preferences of respondents.

| Indicator | Sample Size | Percentage (%) | Percentage in Xi'an (%) [62] |
|---|---|---|---|
| Gender | | | |
| Male | 597 | 46.39 | 51.87 |
| Female | 690 | 53.61 | 48.89 |
| Age (years) | | | |
| ≤18 | 142 | 11.03 | NA |
| 19–45 | 913 | 70.94 | NA |
| 46–65 | 223 | 17.33 | NA |
| >65 | 9 | 0.70 | 11.69 |
| Landscape preferences | | | |
| Open green spaces | 205 | 15.93% | NA |
| Partly open green spaces | 175 | 13.60% | NA |
| Closed green spaces | 98 | 7.61% | NA |
| Partly blue spaces | 737 | 57.26% | NA |
| Partly open squares | 72 | 5.59% | NA |
| Total sample size | 1287 | 100% | NA |

## 3. Results

### 3.1. Sample Description

A total of 1287 valid questionnaires were completed online. There were 690 (53.6%) questionnaires completed by women and 913 (70.9%) by young and middle-aged people. People who lived in medium- or high-risk management areas could not leave their apartments. More than half of the respondents preferred partly open blue spaces (57.26%), followed by open green spaces (15.93%), partly open green spaces (13.6%), closed green spaces (7.61%), and partly open squares (5.59%).

### 3.2. Associations among Houseplants, Window Green Views, and Anxiety

As the logistic regression results show in Table 4, during the lockdown, the number of houseplants ($p < 0.05$) and window green views ($p < 0.01$) had positive contributions on anxiety remission.

As shown in Table 5, an increase in the number of houseplants had a positive effect on relieving mild anxiety. Compared with respondents living with three or more pots of houseplants, the probability of moderate and severe anxiety of respondents who lived with no houseplants was 1.609 times higher.

More green views from windows had a positive effect on relieving mild and moderate anxiety. Compared with respondents living with three trees or a grove outside their windows, the probability of mild anxiety was 2.314 times higher for respondents living without trees outside their windows. The probability of mild anxiety in respondents living with one or two trees outside their windows was 1.946 times that of respondents living with three trees or a grove outside their windows. Similarly, compared with respondents living with three trees or a grove outside their windows, the probability of moderate and severe anxiety in respondents living without trees outside their windows was 2.497 times higher. The probability of moderate and severe anxiety for respondents living with one or

two trees outside their windows was 1.763 times that of respondents living with three trees or a grove outside their windows.

**Table 4.** The *p*-value of indicators in Regression 1.

| Indicator | *p*-Value | *p*-Value (Added Landscape Preferences) |
|---|---|---|
| Gender | 0.928 | 0.838 |
| Age | 0.197 | 0.230 |
| Floor | 0.381 | 0.427 |
| Orientation of the windows | 0.434 | 0.421 |
| Number of houseplants | 0.042 | 0.035 |
| The presence of green space, water body or garden outside the windows | 0.403 | 0.454 |
| Number of trees outside the windows | <0.001 | <0.001 |
| Heard birdsong or not | 0.689 | 0.774 |
| Frequency of looking out of windows | 0.071 | 0.059 |
| Frequency of visiting park before | 0.960 | 0.968 |
| Landscape preferences | NA | 0.046 |

**Table 5.** Associations between UGS exposure and anxiety in Regression 1.

| | Level of Independent Variable | Mild Anxiety | | | Moderate and Severe Anxiety | | |
|---|---|---|---|---|---|---|---|
| | | Exp (B) | CI Lower | CI Upper | Exp (B) | CI Lower | CI Upper |
| Number of houseplants | 0 pot | 1.372 | 0.948 | 1.988 | 1.609 * | 1.008 | 2.567 |
| | 1–2 pots | 0.845 | 0.600 | 1.189 | 0.829 | 0.519 | 1.323 |
| | 3 pots [1] | NA | | | NA | | |
| Number of trees outside the windows | Nearly 0 tree | 2.314 ** | 1.348 | 3.972 | 2.497 * | 1.267 | 4.923 |
| | 1–2 trees | 1.946 ** | 1.412 | 2.682 | 1.763 ** | 1.148 | 2.707 |
| | More than 3 trees, or contiguous growth [1] | NA | | | NA | | |
| Frequency of looking outside windows | Nearly 0 times a day | 0.817 | 0.474 | 1.407 | 1.051 | 0.554 | 1.993 |
| | 1–2 times a day | 0.689 * | 0.513 | 0.925 | 0.690 | 0.465 | 1.023 |
| | More than 3 times a day [1] | NA | | | NA | | |

[1]: reference item, *: $p < 0.05$, **: $p < 0.01$.

### 3.3. Anxiety Remission Effects of Green Exposure among Different Landscape Preference Groups

Our results showed that both houseplants and the visual presence of trees outside windows helped to reduce residents' probability of anxiety during the lockdown period. Their effects varied among respondents with different landscape preferences (Table 6). The effects of the other green exposure variables were not significant.

The presence of houseplants helped reduce the probability of anxiety in residents who preferred open green spaces, partly open green spaces, and partly open blue spaces, but this effect was not obvious in residents who preferred closed green spaces (Table 6). The influence of an increase in houseplants was not significant. Among respondents who preferred open green spaces, the probability of moderate and severe anxiety for respondents who had no houseplants was 6.434 times that of respondents who had three pots of houseplants. Among respondents who preferred partly open green space, the probability of moderate and severe anxiety in respondents who had no houseplants was 4.686 times that of respondents who had three pots of houseplants. In respondents who preferred closed green spaces, and those who preferred partly open blue spaces, the probability of anxiety was not significantly affected by houseplants.

**Table 6.** Associations between UGS exposure and anxiety of different landscape preferences in Regression 2.

| Landscape Preferences | Variable | Level of Independent Variable | Mild Anxiety | | | Moderate and Severe Anxiety | | |
|---|---|---|---|---|---|---|---|---|
| | | | Exp (B) | CI Lower | CI Upper | Exp (B) | CI Lower | CI Upper |
| Open green spaces | Houseplants | 0 pot | 0.815 | 0.228 | 2.580 | 6.434 ** | 1.477 | 28.03 |
| | | 1–2 pots | 0.827 | 0.345 | 1.982 | 1.736 | 0.428 | 7.045 |
| | | 3 pots [1] | NA | | | NA | | |
| | Green views through windows | Nearly 0 tree | 1.196 | 0.201 | 7.124 | 0.314 | 0.028 | 3.418 |
| | | 1–2 trees | 3.818 ** | 1.506 | 9.680 | 0.527 | 0.158 | 1.750 |
| | | More than 3 trees, or contiguous growth [1] | NA | | | NA | | |
| Partly open green spaces | Houseplants | 0 pot | 1.267 | 0.413 | 3.885 | 4.686 * | 1.067 | 20.570 |
| | | 1–2 pots | 0.974 | 0.341 | 2.777 | 1.161 | 0.230 | 5.872 |
| | | 3 pots [1] | NA | | | NA | | |
| | Green views through windows | Nearly 0 tree | 17.251 ** | 2.295 | 129.648 | 10.951 * | 1.051 | 114.068 |
| | | 1–2 trees | 2.211 | 0.866 | 5.647 | 2.010 | 0.553 | 7.294 |
| | | More than 3 trees, or contiguous growth [1] | NA | | | NA | | |
| Closed green spaces | Houseplants | 0 pot | 2.407 | 0.354 | 16.379 | 4.137 | 0.454 | 37.709 |
| | | 1–2 pots | 0.468 | 0.095 | 2.299 | 0.165 | 0.017 | 1.574 |
| | | 3 pots [1] | NA | | | NA | | |
| | Green views through windows | Nearly 0 tree | 0.651 | 0.038 | 11.093 | 13.276 | 0.551 | 319.895 |
| | | 1–2 trees | 3.843 | 0.959 | 15.393 | 0.315 | 0.034 | 2.917 |
| | | More than 3 trees, or contiguous growth [1] | NA | | | NA | | |
| Partly blue spaces | Houseplants | 0 pot | 1.321 | 0.809 | 2.159 | 0.953 | 0.501 | 1.810 |
| | | 1–2 pots | 0.703 | 0.433 | 1.140 | 0.840 | 0.460 | 1.535 |
| | | 3 pots [1] | NA | | | NA | | |
| | Green views through windows | Nearly 0 tree | 3.363 ** | 1.763 | 6.413 | 3.488 * | 1.565 | 7.773 |
| | | 1–2 trees | 1.788 ** | 1.176 | 2.720 | 2.395 ** | 1.385 | 4.143 |
| | | More than 3 trees, or contiguous growth [1] | NA | | | NA | | |

[1]: reference item, *: $p < 0.05$, **: $p < 0.01$.

Through-window green views helped reduce the probability of anxiety in residents with landscape preferences (Table 1), but the effects were different in preference groups (Table 6). Among respondents who preferred open green spaces, the probability of mild anxiety for those living with one or two trees outside their windows was 3.818 times that of respondents living with three trees or a grove outside their windows. In contrast, their probability of experiencing moderate and severe anxiety was not significantly affected by through-window green views.

Among respondents who preferred partly open green spaces, the probability of mild anxiety with no trees outside the window was 17.251 times that of other respondents. The probability of moderate and severe anxiety in respondents living with no trees outside their windows was 10.951 times that of respondents living with three trees or a grove outside their windows.

The respondents who preferred closed green spaces showed a probability of anxiety that was not significantly affected by through-window green views.

In respondents who preferred partly open blue spaces, the probability of mild anxiety with no trees outside their windows was 3.363 times higher than the respondents living with three trees or a grove outside their windows. The probability of mild anxiety in respondents living with one or two trees outside their windows was 1.788 times higher than those with three trees or a grove outside their windows. The probability of moderate and severe anxiety for respondents living with no trees outside the window was 3.488 times higher than those with three trees or a grove outside their windows. The probability of moderate and severe anxiety in respondents living with one or two trees outside their windows was 2.395 times higher than those with three trees or a grove outside their windows.

## 4. Discussion

### 4.1. The Role of Houseplants and Window Green Views during the Lockdown

During the lockdown period, some governments issued orders to restrict access to parks [63]. This created a chance for us to find the role of different green features within and/or surrounding living environments when controlling UGS accessibility and usability. During the two-month lockdown period, people lived indoors, their houseplants and green views from the windows beneficial to relieve their anxiety. This was consistent with the results of long- and short-term residential green space exposure studies [21,64], and with the results of previous studies in lockdown period [27,30,40].

This study found that houseplants and green views from windows had similar anxiety remission effects when public UGS were inaccessible or unavailable during lockdown. This was different from a study under pandemic but not lockdown conditions, which found that home gardens were more important than the anxiety remission effects of green views from windows [28]. The results of this study also showed that even in winter, when about 60% trees have completely lost their leaves, green views through windows still have anxiety remission effects as in other studies in the spring and summer [27,29,39].

In the future, when designing new apartment houses, it should be ensured that each household has a balcony, which can increase the green view and provide the necessary space for house plants. When updating old residential areas, scattered spaces can be used to carry out participatory greening activities, such as pocket gardens built by residents, which can increase green space for residential areas while promoting residents' contact with nature and improving residents' well-being.

### 4.2. The Role of Birdsong during the Lockdown

Different from previous studies [33,34], in this study the anxiety remission effects of birdsong were not obvious. In this study, 28.4% of respondents lived on floors 12 or higher. They may have had difficulty hearing birdsong in surrounding green spaces. Evidence has shown that the higher the floor, the lower the noise level detected from the ground [65]. Furthermore, this study was conducted during the winter rather than the breeding period, with low forest bird densities [66]. The residents' perceived psychological restorative effects

of birds were less than in spring and summer [67]. Finally, UGS are important habitats for urban birds. Bird diversity and UGS size [68] were positively associated with green views and UGS size [23]. So, anxiety remission effects of birdsong were easily obscured by the effects of through-window green views in our regression model.

### 4.3. The Anxiety Remission Effects on Residents with Different Landscape Preferences

Our results showed that both houseplants and green views from the windows had a similar anxiety remission effect. These results were similar to those of previous studies during the lockdown period [27,29]. In addition, the results showed that residents' habits of visiting green spaces or parks, and their frequency of looking out through windows, had no obvious anxiety remission effects during the lockdown. The significance was further decreased by adding the landscape preference variable in the regression mode. This was different from a study conducted during the Italian lockdown [32]. This could be related to the fact that residents' behavior and lifestyle had been changed at the end of the second year of the COVID-19 pandemic [3,4]. Therefore, in future research, it is necessary to consider not only residents' behaviors and habits but also their landscape preferences, which will be conducive to the formation of more targeted strategies.

This study found that residents preferred partly open blue spaces the most (57.26%), followed by open green spaces, partly open green spaces, closed green spaces, and partly open squares. This aligned with the results of Gao (2019), but differed from the studies of Cottet (2018) and Hoyle (2019). This was due to the different landscapes that people prefer in different scenes [46]. In our study, the question was "If possible, which kind of green spaces would you most prefer to see outside your windows?" Therefore, respondents were more inclined to choose open and partly open green spaces. Studies by Cottet (2018) and Hoyle (2019) focused on a natural section of river and public parks, so their respondents were more inclined to choose more natural landscapes.

Different from previous studies that constructed virtual landscapes through eye tracking data [44] and photos [47] to explore the relationship between people's landscape preferences and their anxiety remission effects, this study understood people's landscape preferences by simulating pictures. Based on this, the anxiety remission effects of window green exposure and indoor green on people with different landscape preferences were explored. This study found that house plants had anxiety remission effects on residents who preferred open green spaces and partly open green spaces, but there was no obvious effect on residents who preferred closed green spaces or on residents who preferred partly open blue spaces. The presence of through-window green views had a remission effect on residents who preferred open green spaces and those who preferred partly open green spaces. An increase in through-window green views had a remission effect only on residents who preferred partly open blue spaces. Residents who preferred closed green spaces were not affected by their windows' green views. So, the influence of houseplants and through-window green views was associated not only with people's preference for openness in UGS but also with their preferences for UGS composition. Therefore, in future planning for residential green spaces, considering and increasing tree cover can efficiently benefit a wider range of people in response to future public health crises. Also, in some places where incapacitated people live for a long time, such as nursing homes, the affiliated green space planning can be based on the results of landscape preference surveys, so as to promote users' mental health and well-being.

### 4.4. Limitations

When interpreting our results, insufficient information on spatial heterogeneity of samples is one of the limitations of this study. Because the authors were also in a lockdown state during the questionnaire survey, they could not conduct field works and face-to-face surveys, and the precise residence of the respondents could not be obtained in the online questionnaire. Moreover, it was challenging to analyze the urbanization level of residents in different locations of the city, and variations in building density, green space quantity

and quality, population density, air pollution and noise level and could not link them with anxiety levels by GIS, which limited the reference value of this paper for spatial planning.

According to other studies published on mental health during times of COVID-19, social factors such as education, income, social isolation and house type affect people's mental health [27–29]. Since age and gender factors were considered as demographic variables, the interference effect caused by differences in other socioeconomic attributes of the samples could not be identified.

Other limitations of our study relate to the survey season. Due to the high incidence of epidemic infectious diseases in winter, the survey was limited to the winter. When 60 trees in the study area were withered and the proportion of herbs was higher, the level of exposure to green spaces was reduced. Future studies can conduct a meta-analysis to investigate the contribution of green space exposure and indoor green plants to residents' mental health under different seasons and similar lockdown rules. Despite these limitations, our results additionally understood the contribution of green exposure to residents' mental health during times of COVID-19 from a new perspective.

## 5. Conclusions

Houseplants and UGS exposure through windows had alleviating effects on residents' anxiety during the lockdown period. These effects were associated with people's preferences concerning openness and composition of UGS.

The presence of indoor houseplants was beneficial in cases of moderate and severe anxiety among respondents who preferred open green spaces and partly open green spaces. Nevertheless, the presence of houseplants was not effective for residents who preferred closed green spaces or partly open blue spaces.

Exposure to UGS through windows alleviated mild anxiety in respondents who preferred open green spaces. It also alleviated mild, moderate, and severe anxiety in respondents who preferred partly open green spaces. More UGS exposure from windows alleviated mild, moderate, and severe anxiety in respondents who preferred partly open blue spaces. However, it was not effective for those who preferred closed green spaces.

In future pandemics and potential lockdowns, distributing indoor plants to households can relieve public mental health crises. In addition, enhancing residential areas with more UGS will sustainably improve resilience during public mental health crises. These methods can be useful in some places where incapacitated people live for a long time during non-lockdown periods, such as nursing homes. It is necessary to consider not only residents' habitual uses of UGS, but also their preferences concerning the openness and composition of UGS. Such considerations will make cities healthier, more livable, and more sustainable.

**Author Contributions:** Z.Z. wrote the main manuscript text and prepared tables. H.Z. carried out data collection and prepared Figure 2. H.Y. carried out data collection too. B.Z. carried out grammar checking and reference collation. All authors have read and agreed to the published version of the manuscript.

**Funding:** This research was funded by the Start-up Program for Ph.D Research of Northwest A&F University (No. 2400Z1090121116 (2452021121)), China, the Shaanxi National Science Foundation (No. 2021JQ-175), China.

**Institutional Review Board Statement:** The studies involving human participants were reviewed and approved by the Ethics Committee of the college of Landscape Architecture and Arts, Northwest A&F University. Written informed consent to participate in this study was provided by the participants.

**Informed Consent Statement:** Not applicable.

**Data Availability Statement:** Not applicable.

**Conflicts of Interest:** The authors declare no conflict of interest.

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
