# Peer review of "Home Greenery: Alleviating Anxiety during Lockdowns with Varied Landscape Preferences"

_sustainability, doi:10.3390/su152115371_

Round 1
Reviewer 1 Report
Overall:
This study aimed to assess the psychologically restorative effects of indoor UGS exposure for people with different landscape preferences when public green spaces were inaccessible during a COVID-19 lockdown. In general, the topic is interesting, but the manuscript lacks important information related to the methodology. This section should be improved and supplemented.
Materials and Methods:
· Was the sample of the study participants representative for the analysed city? Maybe the authors could provide comparison of the main characteristics of study participants with the whole population of Xi’an.
· Have you assessed information related to house type? Have you asked whether the person lives in a detached house or an apartment building? Those living in a detached house have more contact with greenery and this could result in different effects.
· The description of demographic variables could be expanded by indicating if variable was continuous or categorical, providing categories, etc. In 2.2.5 subsection, district is mentioned but did not explain in more detail.
· What was the basis to distinguish three anxiety levels?
· In Table 3, age groups are specified with an error, I guess it should be more or equal to 18 and 66.
· Was the tree type (coniferous, deciduous) evaluated during survey? It is very important, especially during winter (as the survey was conducted in January), when deciduous trees are leafless and lose its immediate effect.
· It would be good to include air pollution and noise data when analysing the association between houseplants, green view, and psychological state, because there is mediating effect.
· It is necessary to present information about the closest green space from the residence of participants.
· The map showing geographic location and area where the study was conducted would be useful.
Results:
· What are the confounders that were used for the adjustment in regression (Table 5)?
Discussion
· Discussion should be supplemented with more practical implication of the study results.
· Limitation section or paragraph is missing.
Minor errors:
· UGS abbreviation should be explained.
Some editing for English language is required throughout the manuscript.
Reviewer 2 Report
A customised but interesting article. Raises an important and underestimated aspect of ecotherapy as an important part of a person's quality of life.
Methodologically sound project, discussion adequate and interesting.
A work worthy of publication.
I suggest to add at the end of the discussion the limitations regarding the study.
Author Response
Discussion: add at the end of the discussion the limitations.
Response: Thanks for pointing this out. We added limitations in line 366-387.
Reviewer 3 Report
This is a very interesting study investigating the effects of indoor green exposure on people with different landscape preferences during the COVID-19 lockdown. The paper is well written and of interest for the journal; however, several minor changes are recommended before considering it for publication.
ABSTRACT.
1- How were the participants recruited? This should be explained, briefly, in the abstract section.
2-Is the questionnaire validated and designed by the authors?
3- The conclusions should be drawn irrespective of the occurrence of future pandemics. Why are indoor plants important for mental health?
INTRODUCTION
1- The introduction is well written. I recommend to add some references about the "Blue zones". These would help and reinforce the importance and the impact of green exposure on mental health.
2- PLease, emphasize the comparison between urban green zones and rural zones.
MATERIAL AND METHODS.
1- I recommend to start the methods section with a subsection called "Participants and study design".
RESULTS
1- Previous to the study design, were the authors expecting differences between men and women in the association between houseplants, window green views and the appearance of anxiety symptoms?
DISCUSSION
The impact of urbanicity or urban degree on mental health have been extensively investigated. In fact, urbanicity can be considered a social determinant of mental health; a social factor potentially impacting mental health (also physical health). I recommend to add several references about social determinants of health.
Round 2
Reviewer 1 Report
The authors have addressed all comments and suggestions.
Author Response
Many thanks for your comments.